# An Overview of Macrolide Resistance in Streptococci: Prevalence, Mobile Elements and Dynamics

**DOI:** 10.3390/microorganisms10122316

**Published:** 2022-11-23

**Authors:** Dàmaris Berbel, Aida González-Díaz, Guillem López de Egea, Jordi Càmara, Carmen Ardanuy

**Affiliations:** 1Microbiology Department, Hospital Universitari de Bellvitge, IDIBELL-UB, 08907 Barcelona, Spain; 2Research Network for Respiratory Diseases (CIBERES), ISCIII, 28020 Madrid, Spain; 3Department of Pathology and Experimental Therapeutics, School of Medicine, University of Barcelona, 08007 Barcelona, Spain

**Keywords:** macrolide resistance, *Streptococcus*, *Streptococcus pneumoniae*, *Streptococcus pyogenes*, GAS, pneumococcus, mobile genetic elements, ICE, IME

## Abstract

Streptococcal infections are usually treated with beta-lactam antibiotics, but, in case of allergic patients or reduced antibiotic susceptibility, macrolides and fluoroquinolones are the main alternatives. This work focuses on studying macrolide resistance rates, genetic associated determinants and antibiotic consumption data in Spain, Europe and also on a global scale. Macrolide resistance (MR) determinants, such as ribosomal methylases (*erm*(B), *erm*(TR), *erm*(T)) or active antibiotic efflux pumps and ribosomal protectors (*mef*(A/E)-*mrs*(D)), are differently distributed worldwide and associated with different clonal lineages and mobile genetic elements. MR rates vary together depending on clonal dynamics and on antibiotic consumption applying selective pressure. Among *Streptococcus*, higher MR rates are found in the viridans group, *Streptococcus pneumoniae* and *Streptococcus agalactiae*, and lower MR rates are described in *Streptococcus pyogenes*. When considering different geographic areas, higher resistance rates are usually found in East-Asian countries and milder or lower in the US and Europe. Unfortunately, the availability of data varies also between countries; it is scarce in low- and middle- income countries from Africa and South America. Thus, surveillance studies of macrolide resistance rates and the resistance determinants involved should be promoted to complete global knowledge among macrolide resistance dynamics.

## 1. Streptococcus

The genus *Streptococcus* comprises a large number of species found in humans and animals as part of the normal microbiota. Some of them had successfully adapted to progress from colonization to cause invasive disease, involving different bacterial processes, such as adherence and invasion or resistance to host immune responses [1,2]. While some streptococcal species are almost exclusively restricted to a specific host, such as *Streptococcus suis* in pigs, others can be found in multiple hosts, such as *Streptococcus agalactiae*. Moreover, these streptococci that are not usually part of the human microbiota, such as *Streptococcus suis,* can cause zoonotic infections in the process of animal and human interaction. Then, antibiotic animal policies to treat streptococcal infections may have a direct impact on streptococci causing human infections [1,2].

As human pathogens, streptococci can cause a wide variety of infections, such as pneumonia, meningitis, endocarditis, otitis media, sepsis or skin and soft-tissue infections. Among them, *Streptococcus pneumoniae* and *Streptococcus pyogenes* are major human pathogens; they are an important cause of morbidity and mortality worldwide and are the main representative of the mitis and pyogenic group of streptococci [1,2,3,4,5].

Although streptococcal infections are mainly treated with beta-lactam antibiotics, macrolides are an important group of antibiotics used either as combined therapy in severe infections, such as community-acquired pneumonia, as an alternative therapy in beta-lactam allergic patients or as adjunctive therapy for its protein inhibition effect in toxin-related diseases, such as streptococcal toxic shock syndrome (SSTS). This review provides a synopsis of macrolide resistance in pathogenic *Streptococcus* species regarding rates (Table 1 and Appendix A) and mechanisms of resistance (Table 2), and also the major drivers of resistance: mobile genetic elements (MGE), clonal dynamics and antibiotic consumption [1,2,6,7]. Worldwide macrolide resistance rates are summarized in Table 1 and more extendedly studied in Appendix A.

## 2. Macrolides and Resistance Mechanisms

Macrolides are a group of antibiotics made up of a macrolactone ring, varying in size depending on the carbon atom composition of the lactone ring, with neutral or amino sugar groups attached [6,7]. Macrolides classification is based on the carbon atom composition of the lactone ring: 14-member ring, erythromycin and clarithromycin; 15-member ring, azithromycin; and 16-member ring, spiramycin and josamycin. Erythromycin, naturally produced by *Streptomyces erythreus,* was discovered in 1952. With the exception of spiramycin (produced by *Streptomyces ambofaciens*), the remaining macrolides are semisynthetic, as a result of subsequent modifications to improve the pharmacokinetics and spectrum of activity. Macrolides inhibit peptide chain elongation by reversibly binding to 23S rRNA at the peptidyl-tRNA binding region, blocking the nascent peptide channel exit, inhibiting translocation, and thus peptidyl-tRNA drop off leading to abortion of translation. Smaller macrolides (14-and 15-membered) partially block the channel in a manner that allows only short oligopeptides (6–8 Aa) to form, but 16-membered macrolides fully block the channel and cause ribosomal disassociation [6,7,8]. The emergence of macrolide resistance is a cause for concern and macrolides have been included in the list of critically important antimicrobials for Human Medicine of the World Health Organization (WHO) [9].

Resistance to macrolides in the genus *Streptococcus* is due to three different mechanisms: ribosomal post- and pre-transcriptional modifications, active expulsion of the antibiotic by efflux pumps and target protection [6,8,10,11,12,13]. These resistance mechanisms usually confer resistance to other antibiotic groups which have their target site in the 50S ribosomal unit, such as lincosamides or streptogramins. This fact is the basis of phenotypic classification of macrolide and lincosamide resistance.

The post-transcriptional ribosomal modification is performed by methylases encoded by the *erm* (erythromycin ribosome methylase) genes that mono- or dimethylate the A2058 in the V domain of 23S rRNA [6,10,11]. This methylation causes a conformational change in the ribosomal peptidyl transferase center (PTC) 50S ribosomal subunit, and thus high-level resistance to macrolides, lincosamides and group B streptogramins (MLS_B_ phenotype) that can be constitutively (cMLS_B_) or inducibly (iMLS_B_) expressed [6,10,11]. When inducible, clindamycin may appear susceptible, but, in the presence of a macrolide, resistance rises; for this reason, antibiotic susceptibility testing should include induction tests, such as a screening by disk–diffusion D-zone to avoid false susceptibility results [11]. More than 20 classes of *erm* genes have been identified [7], but in streptococci the most frequently found are *erm*(B), *erm*(TR) (also known as a subclass of *erm*(A)) and *erm*(T) [6,10,11,14,15,16,17,18]. 

Resistance due to ribosomal mutations in the 23S rRNA or L4 and L22 proteins is very rare in streptococci, conferring different resistance phenotypes depending on the mutations found [6]. For instance, mutations in 23S rRNA, or near the macrolide binding residue A2058, result in different levels of macrolide resistance, depending on the copies of the rRNA operons mutated, which means that high-level resistance requires mutation of most of these operons [6,19,20,21]. Regarding riboproteins, substitutions in the L4 protein (*rplD*) such as K63E, deletions in L4, such as Del65WR66 or deletions of the 82ME84 in the L22 (*rplV*) are the most frequently found associated with macrolide resistance [6,20,22,23,24]. 

The active expulsion of the antibiotic is determined by the major facilitator superfamily (MFS) efflux pumps encoded in the *mef* gene (macrolide efflux family) [6,10,11,17,18,22]. This mechanism confers low-level resistance to 14- and 15-membered ring macrolides (erythromycin and azithromycin), while there is no resistance to 16-membered macrolides, lincosamides, and streptogramins producing the M-phenotype [6,10,13]. Several allelic variants or subclasses of the *mef* gene have been described, although they share 90% identity between them: *mef*(A) originally reported in *S. pyogenes*, *mef*(E) later identified in *S. pneumoniae*, *mef*(I) also in pneumococcus, and *mef*(O) in Group A *Streptococcus* (GAS) [6,10,11,14,22,25,26]. The *mef* gene was originally identified as the resistance determinant responsible for type M resistance to macrolides, but *mrs*(D) (formerly known as *mel*) always adjacent to *mef*(A) and *mef*(E) has been recently related to macrolide resistance [27,28,29,30,31]. In this way, the Mef protein is responsible for the expulsion of the antibiotic, while [12,27] Mrs(D), as an ABC-F protein, is involved in antibiotic resistance through target protection by driving dissociation of bound antibiotic molecules from the ribosome, safeguarding ribosomal function [32]. 

The worldwide distribution of the macrolide resistance determinants may vary associated to a specific clonal distribution or a horizontal transmission by mobile genetic elements. Table 2 summarizes the frequency of the macrolide resistance determinants (*erm*(B), *erm*(TR), *erm*(T) and *mef*(A/E)) among different geographical regions.

## 3. Macrolide Resistance in β-Haemolitic *Streptococcus*

The first macrolide-resistant *S. pyogenes* (MR-GAS) was reported in 1968 and, since then, the macrolide resistance rates have varied geographically and temporally, associated with clonal dynamics, outbreaks and antibiotic consumption [33]. In *S. pyogenes*, macrolide resistance rates vary widely depending on geographic areas (Table 1); rates tend to be very low (<4%) in some European countries, mild (5–39%) in other European countries and the United States of America (USA), and higher (>40%) in Asian countries (Figure 1). Unfortunately, epidemiological studies of macrolide-resistant (MR) GAS in Africa and South America are scarce.

In Spain, macrolide-resistant *S. pyogenes* emerged in the early 1990s and the M phenotype became predominant by clones encoding *mef*(A) [17,34,35]. The MR-GAS prevalence rapidly rose, reaching a peak from 26% to 44% during the early 2000s in both invasive and non-invasive isolates. This increase was related to a high prevalence of clones associated with *erm*(B) genes and the MLS_B_ phenotype, such as *emm*11-ST403 or *emm*28-ST52. Geographical variations were also related to outbreaks, as occurred in Barcelona among the intravenous drug use population [17,34,35]. Later, most of the epidemiological studies have shown a reduction in macrolide resistance with rates below 10% after 2005, linked to a decline in prevalence of major MR clones [3,17,36]. The decreasing trend continued in time, with current MR rates in the range of 4% in Northern Spain in 2015, 8.5% in a national surveillance study in 2019, or 11.7% in 2016–2018 in adults from Barcelona (Northeast) [37,38]. A similar situation was described in Portugal with a peak of resistance of around 21% in the early 2000s, associated with both M and MLS_B_ phenotypes (Table 2), followed by a decrease of MR rates to 4.4% in 2013, which has been maintained to date with a progressive disappearance of the M phenotype [16,39,40]. Similar data were found in other Southern European countries, such as Greece with resistance rates from 15 to 25% (2003–2017) or 18% in Italy (2000–2013) [41,42,43,44,45]. Data from Northern European countries, such as Norway, Finland, Germany, The Netherlands, United Kingdom (UK) and Ireland showed macrolide resistance rates below 5% in different periods [46,47,48,49,50]. In Europe, the major clones encoding *mef*(A) were *emm*4-ST39 and *emm*75-ST49 and, after the 2000s, mostly *emm*12-ST36. The *erm*(B) was associated with *emm*28-ST52 at first and later it was found in *emm*11-ST403 lineages, and *erm*(TR) raised recently together with *emm*77-ST63 [3,16,17,34,35,36,39,40,41,42,43,44,45,46,47,49,50,51].

**Table 1 microorganisms-10-02316-t001:** Global distribution of macrolide resistance rates summarized by streptococcal species. This table synthesizes what is described in Appendix A, where the references can be consulted.

		*Streptococcus pneumoniae*			*Streptococcus pyogenes*			*Streptococcus dysgalactiae*			*Streptococcus agalactiae*		
Continent	Region	Period	Mean (%)	Range (%)	Period	Mean (%)	Range (%)	Period	Mean (%)	Range (%)	Period	Mean (%)	Range (%)
Africa	Eastern	2016–2017	33.6	*	2021–2021	6.1	*	-	-	-	2021–2021	20.7	*
	Middle	-	-	-	2012–2013	11.0	*	-	-	-	-	-	-
	Northern	1998–2014	15.6	9.4–31.0	2000–2013	4.6	3.6–5.2	-	-	-	-	-	-
	Western	-	-	-	2004–2012	11.1	*	2012–2013	28.1	*	2018–2018	30.3	*
America	Central	2003–2004	29.0	*	1999–2009	4.6	*	-	-	-	-	-	-
	Northern	1986–2019	23.2	3.5–47.3	1997–2019	16.5	3.9–25.0	1997–2004	19.3	-	1997–2004	52.0	26.6–57.0
	South	2003–2019	26.3	0–85.9	1996–2012	4.0	2.4–15.4	1979–2013	16.5	13.9–26.1	2014–2018	18.6	17.3–25.0
Asia	Eastern	2000–2020	89,6	56.6–100	1998–2019	40.2	1.2–95.0	1993–2016	24.3	10.3–71.4	2003–2021	35.5	29.1–40.7
	Southeastern	2008–2009	49.9	5.3–82.9	-	-	-	-	-	-	-	-	-
	Southern	2008–2013	48.2	17.4–79.0	1986–2017	9.4	2.9–53.0	-	-	-	-	-	-
	Western	2003–2016	42.4	14.0–67.7	1996–2019	8.5	4.3–61.3	1995–2011	12,4	6.0–38.9	-	-	-
Europe	Eastern	1995–2016	25.2	4.1–49.1	2013–2017	13.0	10.5–31.0	2008–2017	21.4	*	2016–2018	27.4	22.0–30.4
	Northern	1996–2018	5.7	0.3–24.2	1993–2016	9.8	1.6–11.9	2000–2006	7.3	5.4–15.1	2000–2018	12.3	3.4–17.5
	Southern	1997–2019	21.4	6.1–49.4	1993–2019	14.8	3.2–46.8	2000–2015	23.2	16.7–31.3	2001–2020	33.4	15.7–26.0
	Western	1992–2016	21.3	2.9–53.1	2003–2013	3.4	1.4–4.0	2005–2010	13.2	5.1–26.4	2005–2019	27.1	5.3–34.1
Oceania	Australia	1999–2017	20.4	13.0–31.0	-	-	-	-	-	-	2016–2019	32.0	*

- No data. * Single study.

In the USA, a study showed 9% of MR in the 1990s was mostly due to isolates with the M phenotype [52]. After that, the MLS_B_ phenotype became predominant and the rate of MR increased to 15% in 2015 [53,54]. More recently, the spread of different epidemic strains changed the epidemiology of MR lineages in the USA: the dominant lineage until 2016 (*emm*11-ST403/*erm*(B)) was replaced by *emm*92-ST82, leading to an increase of MR up to 19% in 2017 [55,56,57]. This was related to the spread of a plasmid containing *erm*(T) among successful lineages such as *emm*92-ST82, *emm*4-ST39 and *emm*77-ST399, which changed the epidemiology of the genetic elements associated with MR (Table 2) [55,56,57]. Lately the spread of an *emm*49-ST433/*erm*(TR) clone, balanced the proportion of *erm*(TR)/*erm*(T) and increased MR rates up to 25% in 2019 [57]. Different studies from South America between 1996 and 2009 showed MR rates below 5% in Chile and Argentina 1996–2009 [58,59,60]. Nevertheless, a study from Brazil reported higher resistance rates (15.4%) from 2008 to 2012 [61]. 

Asian countries have the highest rates of MR worldwide; in China, MR increased from 15% in 2000 to 95% in 2016 mainly associated with the spread of *emm*12-ST36/*erm*(B) and *emm*1-ST28/*erm*(B), which accounted for 80% of MR GAS (Table 2) [62,63]. In Taiwan, MR rates also increased from 18% in 2000–2009 to 58% in 2010–2019 linked to the *emm*12-ST36/*erm*(B) lineage [64]. Similar findings were described in Japan with 40% MR rates in different studies from 2011–2018, but with a higher prevalence of the M phenotype (Table 2) [65,66]. Contrarily, in South Korea, very low resistance rates (<5%) from 2003–2019 were reported [67,68]. In Africa and Middle East countries, the reported resistance rates are below 10%, except for Yemen with 61.3% of MR in 2012 [69,70,71,72].

Other β-haemolitic streptococci as *Streptococcus agalactiae* and *Streptococcus dysgalactiae* subsp *equisimilis* (SDSE) have higher macrolide resistance rates worldwide (Table 1). Probably their ability to colonize and cause disease in humans and animals together with the high antibiotic use in the latter have contributed to these rates [73]. MR rates among SDSE from Europe and the USA were about 19% between 1997 and 2006, but higher MR rates have been described in later periods, such as 26.4% in France (2006–2010), 31% in Italy (2000–2010) and 21.4% in Southern Hungary (2017–2018) [52,74,75,76,77]. Asian countries showed the highest MR rates, as much as 71.4% in China (2007–2015), 42% in Korea (2012–2016) and 45.4% in Southern Taiwan (1993–2010), but milder in Central Taiwan (24% in 2007–2011) and Japan (18.4% in 2010–2013) [78,79,80,81,82]. Little studies report MR in Africa and the Middle East; there is a report from Gabon of 28.1% MR rate in 2013 [83] and 8.6% MR in Israel in the 2007–2011 period [84]. Overall, the MLS_B_ phenotype is the most frequently described among SDSE associated with both, *erm*(B) and *erm*(TR) (Table 2). Nevertheless, some exceptions could be found, such as in Taiwan when *mef*(A) accounted for more than a half of MR SDSE isolates [80,81]. Moreover, different lineages are associated with macrolide resistance in SDSE without a specific lineage trend [76,79,81,82,85]. 

Worldwide, macrolide resistant rates are also high in *S. agalactiae*. As occurred with other streptococci, Asian countries have the highest resistance prevalence, up to 70% in China in 2015 and 36% in Korea between 2018 and 2021 [86,87]. Nevertheless, high resistance rates were reported in the USA (55% of MR in the last decade) and some European countries with rates ranging from 17.5% in Denmark (2018) to 20–35% in Spain, Portugal, France and Belgium in 2019, and up to 48% in Italy in 2019 (Table 1) [88,89,90,91,92,93,94,95]. These MR rates are associated with a dominance of the MLS_B_ phenotype related to both *erm*(B) and *erm*(TR) and associated with serotypes III, V, Ib and Ia (Table 2) [86,87,88,89,92,93,94,95,96,97]. Recently, an increase of macrolide resistance has been described in Portugal with rates of up to 35% [94] linked with the spread of a recombinant lineage of clonal complex (CC) 1 expressing type Ib serotype. Surveillance studies regarding MR among β-haemolytic streptococci, along with resistance determinants and associated clones, are needed in order to assess the correct empiric treatments for severe invasive diseases and also vaccine development.

**Table 2 microorganisms-10-02316-t002:** Frequency of macrolide resistance determinants among different geographical regions.

**(a) *Streptococcus pneumoniae***			***erm*(B)**		***erm*(TR)**		***erm*(T)**		***mef*(A/E)**		***erm*(B) + *mef*(A/E)**		
Continent	Period	MR (N)	Mean (%)	Range (%)	Mean (%)	Range (%)	Mean (%)	Range (%)	Mean (%)	Range (%)	Mean (%)	Range (%)	References
Africa	2003–2015	579	46.7	36.0–90.2	-	-	-	-	18.40	6.5–62.5	32.8	0.0–46.4	[98,99,100,101]
America	1995–2012	4070	24.4	9.3–73.7	-	-	-	-	67.90	15.8–81.0	5.9	0.0–52.1	[102,103,104,105,106,107]
Asia	2003–2013	1813	53.6	37.1–76.5	-	-	-	-	26.30	2.0–46.0	12.9	2.0–62.9	[103,108,109,110,111]
Europe	1995–2017	2525	53.0	20.8–93.5	-	-	-	-	67.3	1.7–75.0	6.1	0.0–17.9	[103,112,113,114,115,116,117,118]
Oceania	2003–2005	139	28.8	18.9–32.3	-	-	-	-	36.00	27.4–59.5	34.6	21.6–39.2	[103,119]
**(b) *Streptococcus pyogenes***			***erm*(B)**		***erm*(TR)**		***erm*(T)**		***mef*(A/E)**		***erm*(B) + *mef*(A/E)**		
Continent	Period	MR (N)	Mean (%)	Range (%)	Mean (%)	Range (%)	Mean (%)	Range (%)	Mean (%)	Range (%)	Mean (%)	Range (%)	References
Africa	2000–2013	84	19.0	7.6–57.1	26.1	0.0–38.8	-	-	48.8	14.2–57.6	-	-	[69,120]
America	1996–2017	1128	25.6	4.1–25.7	30.5	0.0–36.9	25.6	0.0–55.0	13.7	0.0–95.6	-	-	[15,55,58,59,60,61,121,122]
Asia	1993–2019	877	54.2	18.6–97.7	3.6	0.0–10.5	-	-	39.7	0.0–80.0	-	-	[62,63,64,66,123,124]
Europe	1995–2016	2873	31.5	5.8–70.9	14.6	0.0–59.0	-	-	40.7	0.0–80.0	-	-	[3,16,17,35,36,39,40,41,43,44,45,47,49,50,51,125,126,127,128]
**(c) *Streptococcus dys s. equisimilis***			***erm*(B)**		***erm*(TR)**		***erm*(T)**		***mef*(A/E)**		***erm*(B) + *mef*(A/E)**		
Continent	Period	MR (N)	Mean (%)	Range (%)	Mean (%)	Range (%)	Mean (%)	Range (%)	Mean (%)	Range (%)	Mean (%)	Range (%)	References
America	1979–2013	30	26.6	12.5–33.3	40.0	25.0–50.0	-	-	33.3	16.7–62.5	-	-	[85,129,130]
Asia	1993–2015	320	31.2	8.5–86.3	9.0	0.0–66.6	-	-	2.8	0.0–64.2	-	-	[62,68,79,80,81,82,131,132]
Europe	2000–2015	154	16.2	5.8–29.1	59.0	35.7–8.7	0.6	0.0–5.8	17.5	0.0–23.5	-	-	[35,49,51,74,75,76]
**(d) *Streptococcus agalactiae***			***erm*(B)**		***erm*(TR)**		***erm*(T)**		***mef*(A/E)**		***erm*(B) + *mef*(A/E)**		
Continent	Period	MR (N)	Mean (%)	Range (%)	Mean (%)	Range (%)	Mean (%)	Range (%)	Mean (%)	Range (%)	Mean (%)	Range (%)	References
Africa	2018–2018	13 *	7.6	-	53.8	-	-	-	23.0	-	-	-	[133]
America	2008–2018	5652	35.0	34.9–64.5	34.0	12.5–70.9	3.20	0.0–16.1	26.9	22.2–54.8	-	-	[88,89,134,135]
Asia	2003–2019	94	70.2	59.3–82.5	22.3	7.5–36.3	-	-	3.1	0.0–5.0	-	-	[68,78,136]
Europe	2000–2019	1219	64.1	35.8–80.9	16.6	0.0–36.6	0.40	0.0–1.0	13.80	2.3–22.0	-	-	[35,46,51,94,95,96,97,137,138]
Oceania	1999–2017	32 *	37.5	-	37.5	-	6.20	-	0.0	-	-	-	[139]

N MR: Number of macrolide-resistant isolates accumulated. * No range of percentages are specified if only one study was found per geographical area per microorganism.

## 4. Macrolide Resistance in *S. pneumoniae*

The prevalence of macrolide resistance in pneumococcus differs depending on numerous factors, such as geographic location (Figure 1, Table 1), selective pressure mediated by antibiotic consumption or the introduction of conjugate vaccines [22]. The first MR *S. pneumoniae* isolates were detected in 1967 in Canada [140], but resistance rates remained low worldwide during the 1970s. The introduction of long-action macrolides resulted in a rapid increase in resistance in many countries [4]. In pneumococci, macrolide resistance is mainly related to the presence of *erm*(B) and/or *mef*(A/E) genes (Table 2) [98,102,108,112,117,118].

Many studies on macrolide resistance have been conducted throughout Europe over the years (Appendix A). In the eastern countries, different trends have been observed. In Bulgaria, the rate remained stable between 1995–2005 (18.9%) and 2006–2010 (19.0%) [113], and an increase to 43.9% was observed in 2011–2016 [141]. Contrarily, in Hungary, resistance apparently decreased from 43.6% in 2003–2004 [103] to 25.3% in 2015–2016 [142]. In some areas of northern countries, the resistance rate was very low, as demonstrated in a study in the Faroe Islands, Denmark, between 2009 and 2011 where only 1.5% of isolates from the children’s nasopharynx were resistant [143], or in Skåne, Sweden, where only 8 resistant isolates of 2131 non-invasive pneumococci (0.3%) collected in the 2016–2018 period were found [144]. In the south, the resistance rates remain at around 20% nowadays. In a study conducted in Croatia during 2005 to 2019, 23.0% of pneumococcus causing invasive disease in adults were non-susceptible to macrolides [145]. Lower rates were found in Slovenia, where the resistant rates in invasive disease between 1997 and 2017 were 15.0% [118]. In Spain, a decrease of macrolide resistance was observed between the 2012–2013 and 2015–2016 periods (from 24.0% to 16.3%) among invasive isolates from adults [5]. 

Throughout the 1990s, the spread of multidrug-resistant clones, such as Spain^23F^-1 or Spain^6B^-2 having *erm*(B), or England^14^-9 harboring *mef*(A), were related to macrolide resistance in European countries [4,112]. Throughout the 2000s, with the introduction of the 7-valent pneumococcal conjugate vaccine (PCV7), these multidrug-resistant clones almost disappeared, but macrolide resistance rates remained high, associated with different lineages. For instance, isolates of CC230 expressing serotypes 19A and 24F and harbouring *erm*(B) have been prevalent in Europe, while the CC320^19A^ clone, harboring both *erm*(B) and *mef*(E), was worldwide disseminated [4]. Besides these, other lineages are also contributing to macrolide resistance. In this way, CC63^15A^, CC558^35B^ and CC386^6C^ harboring *erm*(B) or CC100^33F^ harboring *mef*(E) have been found in several European countries [146,147,148,149,150].

Differences between countries are also described in South America. Before PCV7 introduction, resistance rates in Argentina were 12.1% [103], whereas a study carried out by Zintgraff et al. describes a progressive increase from 20.4% in 2006–2008 to 35.2% in 2017–2018 among invasive disease in children [151]. These results are in concordance with a study performed in Bogota, Colombia, where MR rates in children with invasive disease increased from 4.8% in 2008–2011 to 35.2% in 2014–2019 [152]. In Lima, Peru, the resistance rates were higher, at 85.9% in the 2016–2019 period [153]. On the other hand, in Canada (North America), 19.2% of *S. pneumoniae* isolates from invasive and 22.8% from non-invasive sources were resistant (2007–2016) [154].

In the USA, a recent study of ABCs showed a 29.2% rate of resistance among 2881 invasive pneumococci from 2017 [155]. Among them, around 70% had *mef*(E)-*mrs*(D), mainly associated with CC433^22F^, CC538^35B^ and CC100^33F^ lineages. On the other hand, nearly 30% had the *erm*(B) gene alone (CC63^15A^) or associated with *mef*(E)-*mrs*(D) in serotype 19A isolates (CC320). These results were comparable to a previous study performed with 2316 IPD isolates collected in 2015 [156]. Nevertheless, as occurred in other reports, resistance rates were higher among respiratory samples, reaching 47.3% in a national study conducted in adults in 2018–2019, while resistance among isolates from bacteraemia was 29.6% [157].

The highest MR rates are reported from the Asian continent, most notably in the east. In China, 81.6% of pneumococcus isolates between 2003 and 2004 showed macrolide resistance [103]; from 2008 to 2019 rates increased to more than 95% regardless of age [158,159,160,161], type of disease [162] or carriage (Table 1) [109,160,161]. According to a colonization study in children performed in the city of Chongqing in 2020, the MR rate was 56.6%, which is the lowest rate reported in this country [163]. A slightly lower rate was described in a national study conducted in Japan between 2001 and 2015, where the rate was higher in children (84.1%) than adults (75.3%) [164], and in a study from Chungnam, South Korea, where 79.2% of pneumococci showed MR [165]. 

An eleven-year (1998–2014) surveillance study in Casablanca, Morocco, showed low macrolide resistance rates in all periods, despite a slight increase observed from 9.4% in 1998–2001 to 14.0% in 2007–2014 [166]. A recent study conducted in Addis Ababa, Ethiopia, between 2016 and 2017 found that 33.6% of pneumococci were resistant to macrolides [167]. Similar data were described in a national study in Algeria with a resistance rate of 31.0% between 2001 and 2010 [168]. Pneumococcus causing disease in immunocompromised patients in Tunis, Tunisia (2005–2011) presented higher resistance rates with 69.5% of isolates non-susceptible to this antibiotic family [169]. In Australia, 14.8% of *S. pneumoniae* isolated in the children’s nasopharynx (1999–2005) were non-susceptible to macrolides [170], the rate being higher in invasive isolates collected in 2005 (31.0%) [119]. A recent study (2011–2017) showed a reduction in this rate (13.0%) and also in invasive disease in children [171].

## 5. Macrolide Resistance in Other Streptococci

Viridans group streptococci (VGS) belong to the microbiota from the upper respiratory tract, gastrointestinal tract and female genital tract [172]. As commensal microorganisms, they have a low pathogenic potential in immunocompetent individuals, but can cause invasive disease (endocarditis, pneumonia, intra-abdominal infection and shock) [173]. An increasing problem associated with this group is that they can act as a reservoir of resistance genes [174]. It has been demonstrated that bacteria living in biofilms in areas such as the throat readily share genes with each other [175]. For example, some macrolide-resistance genes can be transferred from commensal VGS to pathogenic *S. pyogenes* or *S. pneumoniae* [176,177]. There is concern about the antimicrobial resistance of VGS because of the treatment of its infections, but also because of their role as reservoirs of drug resistance, including mobile genetic elements carrying macrolide resistance determinants [172]. In this way, several studies showed high MR rates among VGS [172,178,179,180]. 

Focusing on macrolides, variable data regarding the resistance mechanisms exist. Some studies found a MLS_B_ phenotype predominance with the *erm*(B) gene [178,181] and others found a M phenotype predominance with the *mef* genes [172,177,180,182].

There are few studies about MR in VGS causing disease that showed an increase in resistance rates over the last three decades as well as changes in the associated phenotypes. In this way, a study of VGS collected from blood cultures in Turkey (1996–2004) found a low resistance rate (27%) with a predominance of the MLS_B_ phenotype due to *erm*(B) [179]. In 2010–2012 a study in Italy about oral VGS in patients with pharyngitis found 56.3% of macrolide resistance, mainly associated with the M phenotype (75%). The most important species carrying this resistance were *Streptococcus mitis*, *Streptococcus oralis*, *Streptococcus parasanguinis*, *Streptococcus sanguinis* and *Streptococcus salivarius*. The most common resistance element found was the macrolide efflux genetic assembly (MEGA) element with *mef*(E) [172]. In line with this, a multicenter study from 12 European countries in 2010–2013 found a 41% macrolide resistance rate among VGS, and 69% of the MR strains showed the M phenotype and 31% the MLS_B_ phenotype [183]. Other non-European studies reported high macrolide resistance rates [148] in VGS collected from sterile sources (Korea 34%, USA 41%, Canada 38% and Northern Taiwan 40%) [178,184,185]. Finally, a study from the USA showed an increase of macrolide resistance between 2010 and 2020 [186]. Other studies explored MR rates and associated genes in VGS carried by healthy people. For instance, a Greek study in healthy children in the late 1990s found 38.5% of macrolide resistance with predominance of the M phenotype (74%), and *S. oralis* showing the highest MR rates [187]. A Finnish study in the elderly found 22.4% of macrolide resistance and 80.6% of the resistant strains had the M phenotype (*mef*(A) gene) [188]. A French study of clinical and commensal *S. salivarius* found 56% and 76% of MR strains, respectively. The predominant phenotype in both samples was the M phenotype [189]. In Belgium, 71% of volunteers between the ages of 17 and 25 years old carried MR VGS with predominance of the MLS_B_ phenotype, while another study in Spain with MR strains from 172 patients showed that the M phenotype with the *mef*(E) gene was predominant in the early 2000s [181,190]. To sum up, MR is highly spread in VGS, as it is shown in different studies around the world. MR rates oscillated between 22% and 76%, but it could be considered a high rate in all samples, age groups and health conditions. The predominant macrolide resistance phenotype was different among the studies, but the M phenotype was explained by the *mef*(A/E) genes and the MLS_B_ phenotype by the *erm*(B) gene in all of them. As these genes could be transferred in MGEs to another bacterial species [14], VGS constitute a resistance reservoir that could endanger the treatment of infections caused by pathogens such as *S. pyogenes* and *S. pneumoniae* [174,176,177].

## 6. Genetic Mobile Elements Related to Macrolide Resistance

Mobile genetic elements, such as integrative and conjugative elements (ICEs), integrative and mobilizable elements (IMEs), plasmids and bacteriophages, are vectors for the transmission of antibiotic resistance in streptococci; they are able to disseminate intra- and interspecies [14,191]. Among them, integrative conjugative elements (ICEs) had all the genes required for conjugation providing the capacity for self-mobilization, while IME do not have all the genes required for conjugation and need the coexistence of an ICE or a conjugative plasmid in the same cell to spread among other streptococci. Most of these elements are characterized by their ability to recombine and to integrate new resistance genes, such as tetracycline, chloramphenicol or aminoglycosides resistance determinants, conferring a multidrug-resistant phenotype. In this way, the role of these MGEs in the dissemination of resistance genes is an important threat to the global emergence of antibiotic resistance in streptococci [14,191,192]. 

Two main *mef* genes (*mef*(A) and *mef(*E)) are found in streptococci. Among them, the *mef*(A) gene is carried into the phage φ1207.3 and is most frequently found in *S. pyogenes.* In this sense, the *mef*(A)-*msr*(D) tandem is found as a necessary signature for the production of an active efflux transport system. Nevertheless, the Tn*1207.1* was associated with the spread of the *mef*(A) gene in pneumococci linked to the international clone England^14^-ST9. On the other hand, the *mef*(E) gene is carried by the MEGA as a *mef*(E)-*mrs*(D) tandem that could also be integrated into larger structures containing other resistance determinants [3,14,192].

Several MGEs have been identified carrying the *erm*(TR) gene, mainly in *S. pyogenes*. Among them, the IME*Sp*2907 and the so-called *erm*(TR)-element are the principal carrier IMEs with different rearrangements and also inserted in larger structures containing other resistance determinants. In this way, the IME*Sp*2907 could be present in the ICE*Sp2905* also carrying *tet*(O), ICE*SpHKU165* structures carrying the *tet*(M) gene as integration of a Tn*916*, or ICE*SagTR7* also carrying *tet*(M) [3,193]. The latter has been found in a mosaic ICE (ICE*Sag236*), which emerged after recombination with the ICE*Spn529IQ*, carrying *mef*(I), and *catQ* and conferring a multidrug-resistant phenotype to *S. agalactiae* [194]. On the other hand, the *erm*(TR)-element has been described inside the ICESp*1108*-like structures that frequently carry *tet*(M) and sporadically *tet*(T). These ICE*Sp1108*-like structures have also been described in *S. agalactiae* and *S. suis*, demonstrating its ability to spread among different streptococci [3,14,193,195].

The *erm*(B) gene is frequently present in macrolide-resistant strains from different streptococcal species, such as *S. pneumoniae*, *S. pyogenes* or *S. agalactiae*. The most important group of elements carrying this gene are transposons of the Tn*916*-family that are characterized by the presence of *tet*(M) [14,192]. Among them, Tn*6002* originated from the insertion of *erm*(B) into the Tn*916* and Tn*6003*, which had a further insertion of the macrolide–aminoglycoside–streptothricin (MAS) element with two *erm*(B) genes; these are the most frequent. On the other hand, the insertion of Tn*917* carrying *erm*(B) into a Tn*916* emerged as the Tn*3872.* These elements are usually located within a Tn*5252*-like structures that frequently carry *cat* gene conferring a multidrug-resistance phenotype. Recently, a new structure (ICE*Sp1070HUB*) carrying *erm*(B) has been described in *S. pyogenes*. This structure also harbors other resistance determinants (*tet*(M), *dfrF*, *cat*(pC194) and the aminoglycoside-modifying enzymes cluster *aph*(3′)III-*sat4*-*ant*(6)Ia) conferring a multidrug-resistant phenotype [3].

The association of *erm*(B) and *mef*(E) genes was related to the spread of dual-macrolide resistance in pneumococci linked to the spread of the CC320^19A^ clone. This dual resistance is carried by the Tn*2010*, which emerged after two recombination events. Firstly, the MEGA element was integrated into Tn*916* becoming the Tn*2009* that further acquired the MAS element carrying the *erm*(B) gene [14]. 

Although the *erm*(T) is sporadically detected in streptococci, its frequency is slightly growing especially in *S. pyogenes*, SDSE and *S. agalactiae*. In these species, the resistance gene was located on small non-self-transmissible plasmids that could be transferred into *trans* profiting from ICE*Sde3396*-like structures [196]. 

## 7. Macrolide Consumption

The emergence and spread of antibiotic resistance is closely associated with antibiotic use. Antibiotics act as a selection pressure that favors the spread of resistant bacteria, either by promoting the spread of pre-existing resistant bacteria or by selecting resistance acquired during the course of treatment. In fact, macrolide consumption has been directly associated with the prevalence of macrolide-resistant *S. pneumoniae* [197], and this may have important clinical consequences [198]. It also seems that macrolide resistance associated with previous exposure could last longer than that caused by other antimicrobial groups [199]. In the same line, experiences with mass drug administration of azithromycin for trachoma control show that this has a direct impact on the increase of macrolide-resistant *S. pneumoniae* carriage [200,201]. Therefore, in addition to the study of the clonal composition of bacteria and the molecular basis of antimicrobial resistance, the study of antimicrobial consumption is also key to understanding the dynamics of antimicrobial resistance.

The consumption of antimicrobials for human use increased worldwide during the 2000–2010 period, mainly due to a strong increase in countries with rapid economic expansion, such as India or China [202]. Despite regional differences, macrolide antibiotics were the third most frequent class sold in 2010, up slightly from 2000. In Europe, macrolide consumption for human use in the community has remained stable over the last years [203]. For example, macrolide consumption changed from 3.2 to 2.9 DDD (defined daily dose) per 1000 inhabitants per day in Spain over the period of 1997–2017. This stability in macrolide consumption has been accompanied by a substantial change in the type of macrolide prescribed. The consumption of short-acting macrolides (i.e., erythromycin) has decreased dramatically (from 0.89 to 0.13 DDD per 1000 inhabitants per day), while the consumption of long-acting macrolides (i.e., azithromycin) has increased (0.55 to 2.03 DDD per 1000 inhabitants per day). In the USA, data on macrolide consumption showed a decreasing trend over the past 10 years, but also with a clear predominance of azithromycin as the most prescribed drug (190 and 90 macrolide prescriptions per 1000 persons and 170 and 86 azithromycin prescriptions per 1000 persons in 2011 and 2021, respectively) [204,205]. In China, the consumption of macrolides in hospitals increased slightly over the period of 2011–2015 (from 1.19 to 1.41 DDDs per 1000 inhabitants per day) [206] with a predominance of roxithromycin (intermediate-acting), clarithromycin and azithromycin as the most used macrolides in 2017 [207]. It should be noted that long-term exposure to azithromycin or erythromycin has been reported to increase the proportion of macrolide resistance in oropharyngeal streptococci [208]. In this study, the proportion of macrolide-resistant streptococci remained higher in patients that received azithromycin when compared to patients that received erythromycin after antibiotic cessation. Then, it is plausible that the pattern of antibiotic replacement from short- to long-acting macrolides has had a direct impact on the selection of macrolide resistance. 

Besides human use, antimicrobials are extensively used in food-producing animals, which also supposes a risk for selecting antimicrobial resistance in bacteria [209]. Moreover, food-producing animals can be a source of human infections caused by resistant bacteria [210]. Macrolide antibiotics have been categorized among the highest-priority critically important antimicrobials by the WHO, requiring measures to reduce the risk of transmission of resistance to humans [211]. The EMA-AMEG (European Medicines Agency, Antimicrobial Advice ad hoc Expert Group) classified macrolides in category C (“caution”) as there are some indications in veterinary medicine with few or no alternatives to macrolides [212]. Global consumption of antimicrobials in food-producing animals has been estimated at 63,151 tonnes in 2010 and is expected to increase to 105,596 tonnes in 2030 due to increased numbers of raised animals [213]. In 2017, China was the largest antimicrobial consumer in food-producing animals, accounting for 45% of global consumption [214]. Despite these estimations, declines in antimicrobial sales have been reported in high-income countries in recent years. Data on antimicrobial consumption in 29 European countries in 2017 showed that the total amount of antimicrobials consumed was superior in food-producing animals than in humans (6558 vs. 4122 tonnes). When adjusted per Kg of estimated biomass, macrolides were the third and the second most frequent group of antimicrobials used among food-producing animals (median 5.7 mg/kg estimated biomass) and humans (median 6.4 mg/kg estimated biomass), respectively [215]. In 2020, macrolide consumption accounted for 8.8% of all veterinary antimicrobial prescriptions in EU/EEA countries and, more importantly, a clear downward trend (mg/PCU) was evident for most antimicrobials, including macrolides, over the last nine years (2011–2020) [216]. In the USA, macrolides were the third most frequent group of medically important antimicrobials marketed in 2020 (7%, total amount of 433,394 kg) and showed a decreasing trend since 2011 (−26% compared to 2020), similar to what occurred with other antimicrobial groups [217]. All these data evidence the effectiveness of the measures adopted to reduce the consumption of antimicrobials in food-producing animals in some countries [218]. It should be noted that interventions to restrict antibiotic use in food-producing animals have been associated with a decrease of antibiotic-resistant bacteria in animals [219]. In an ecological study, a positive association between the use of macrolides in food-producing animals and macrolide resistance in *S. pneumoniae* isolates of human origin has been reported; although this should be interpreted with caution, taking into account all the limitations of this type of study [218]. It has also been described that exposure of soil bacteria to a high macrolide concentration can alter the resistome composition, leading to an increased number of resistance genes to multiple antibiotic classes [48]. Therefore, measures to limit the consumption of medically important antimicrobials, such as macrolides, in food-producing animals are essential to prevent the emergence and dissemination of resistant bacteria.

## 8. Conclusions

In conclusion, macrolide resistance in streptococci varies between countries and species. Over the last decades, surveillance of resistance rates and the genetic studies have helped us to understand the dynamics of resistance among species. In this way, several aspects contribute to the spread of resistance including the dissemination of mobile genetic elements, the antibiotic pressure exerted by consumption in humans and animals, the impact of the introduction of vaccines and the spread of resistant lineages. The dissemination of MGEs carrying resistance determinants among different streptococcal species is a cause for concern and deserves further surveillance. Furthermore, an in-depth analysis of reservoirs of resistance, including non-pathogenic streptococci as well as those causing disease in animals, could be an important step to improve the understanding of resistance dissemination.

## Figures and Tables

**Figure 1 microorganisms-10-02316-f001:**
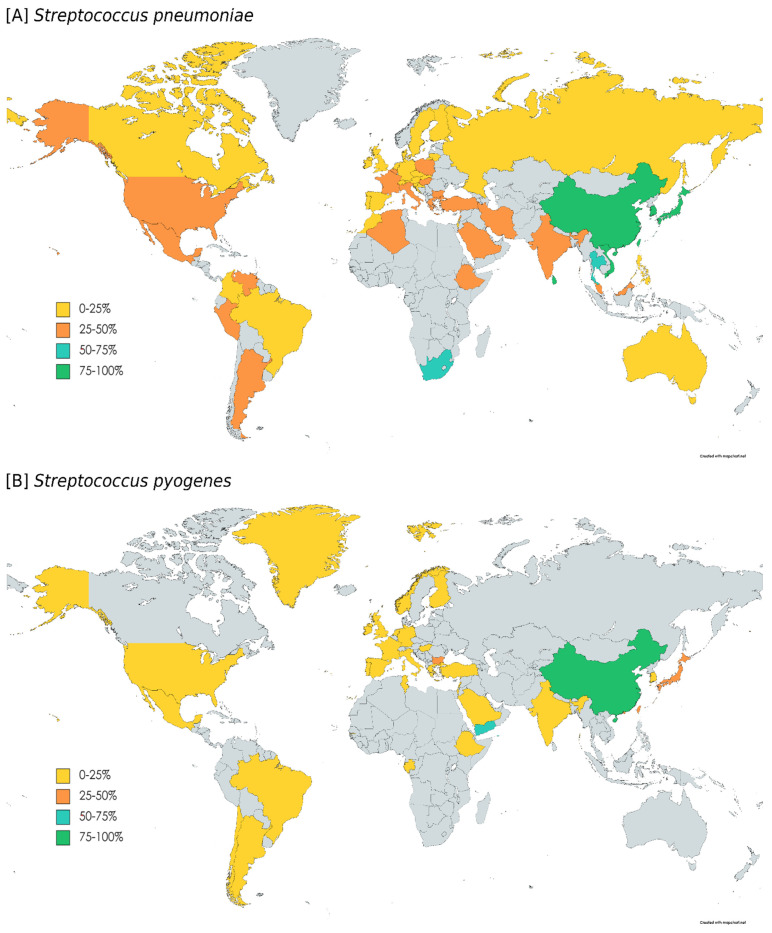
Macrolide resistance rates in the world: *Streptococcus pneumoniae* (**A**) and *Streptococcus pyogenes* (**B**). Macrolide resistance rates reported in Appendix A are depicted as yellow (0–25%), orange (25–50%), blue (50–75%) and green (75–100%) zones, and grey zones mean countries with no data.

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
