# Peer review of "An Overview of Macrolide Resistance in Streptococci: Prevalence, Mobile Elements and Dynamics"

_microorganisms, 2022, doi:10.3390/microorganisms10122316_

Round 1
Reviewer 1 Report
Macrolides are a very important class of antimicrobials, widely used in the treatment of streptococcal infections. The review is interesting because it includes macrolide resistance among different streptococcal species, which is not very common. The authors used a very extensive list of references, aiming at providing a global view of macrolide resistance worldwide.
However, from my point of view, there are some improvements to be made and the first is related to the choice of the papers to include in the review. Table S1 contains data from several countries, but the criteria used to include or exclude papers from the table is not clear. It would be better if the authors explained how was the search performed, detailing keywords, range of date of publication or minimum number of isolates.
Besides this, the table S1 (the basis for table 1) needs to be strongly revised. I did not check all the studies included, but in some, there are some aspects that need clarification and there are a few inaccuracies:
1. One is related to the origin of the isolates – Sanson et al. 2019 (S. pyogenes spreadsheet, lines 18-21) assigned three different disease categories: invasive, skin and soft tissue (SSTI) infections and pharyngitis. In the review the authors considered SSTI as invasive infections, which is inaccurate. The same with the paper from Pato et al, 2018 (S. pyogenes spreadsheet, line 74), which includes SSTI isolates that the authors considered to be invasive.
2. When a study provides data from a given time period and then additionally provides data in smaller periods, the authors should indicate one or the other, otherwise the table will be too big, without adding information. Examples of this are Sanson et al. 2019, Tsai et al. 2021, or Genovese et al. 2020.
3. For many countries, the authors use the review from Farrell et al. 2008. This review includes macrolide resistance data in S. pneumoniae from several countries, but the number of isolates of each country can be small. If there are no studies reporting data from that country, it is good option; in some cases it is hard to understand this choice. Additionally, in some circumstances, there is a good number of papers to provide national data (Germany or Spain), so there is no need to also use the review from Farrell.
4. Some of the country names misspelled (Italy and Finland, for example)
5. In the S. pyogenes spreadsheet, line 7, the only study including 193 isolates in Rafei et al. 2020 is from Lebanon and not Tunisia. In the same spreadsheet, lines 72 and 73 all the isolates are from 2000-2005 (invasive and pharyngeal)
6. In the S. pneumoniae spreadsheet, lines 7-10: the paper from Benbachir et al. 2012 includes an analysis of antimicrobial resistance rates from 1998 to 2008. It does not include strains isolated between 2007 and 2014, so line 10 is not related to this paper. In line 21: respiratory isolates were from 2018-2019 and not 2019-2019
7. Papers including animal and human hosts should be not be used (Silva et al. 2015 has a convenience collection of 115 isolates, but some are of animal origin) – SDSE spreadsheet line 8
8. Again in the S. pyogenes spreadsheet, line 75, the paper cited is a review; the paper the authors should be citing (Declining macrolide resistance in Streptococcus pyogenes in Portugal (2007-2013) was accompanied by continuous clonal changes) includes 3,364 isolates, but resistance rate is 4.4% and not 8.8%.
I did not check every paper, so it is possible that there are more inaccuracies, so it is very important that the authors perform a strong revision of the tables.
Also, there are some references missing from the bibliography.
In the introduction, line 34, the authors stated that “While some streptococcal species are almost exclusively restricted to a specific host, such as Streptococcus suis in pigs, others can be found in multiple hosts like Streptococcus agalactiae. Moreover, these streptococci that are not usually part of the human microbiota can cause zoonotic infections in the process of animal and human interaction”. The specific host the authors are referring to is an animal host? If so, the sentence should be corrected. However the statement that “these streptococci that are not usually part of the human microbiota” does not apply to S. agalactiae (as recognized by the authors in line 180), so this part of the text needs to be altered.
In table 2, I am not able to find the meaning of MD (N). Also, some of the values have an asterisk in superscript that I cannot find in the legend.
Line 213: What is the reference for the description of the first macrolide resistant isolate in 1967?
The description of the genetic lineages of macrolide resistant S. pneumoniae should be updated: besides 19A and 24F, other serotypes emerged in the last years among resistant isolates: 15A, 19F, 23F, 14, 6B, 6C or 33F, to name a few. Authors should look for more references other than 145 and 146.
Figure 1: The different shades of green are difficult to distinguish, so the authors should try to change the colors
Line 388: Please replace macrolides by macrolide
In the section 7, the authors explored well the effect of high macrolide consumption on high levels of resistance; however, there are some countries where resistance is low, despite high consumption of antimicrobials. Examples of this include Greece or Portugal, where decreasing trends in macrolide resistance in GAS were reported, in a situation of high antimicrobial consumption. On the other hand, there were countries where a decrease in macrolide consumption did not lead to a decrease in resistance, and an increase in the proportion of resistant isolates in GAS was detected (for all see Silva-Costa et al., Macrolide-resistant Streptococcus pyogenes: prevalence and treatment strategies). In these cases, alterations in the clonal composition of the population were probably responsible for this different dynamics and this should be further explored and discussed in this paper, for GAS, but also for other species.
Also, analysis of table 1 suggests that in the same geographical region and time period, macrolide resistance can vary from one streptococcal species to another: in Eastern Africa macrolide resistance in GAS was 2% in 2021-2021 and in the same period, resistance among GBS isolates was 20%; in southern Europe, from 1995-2016 pneumococcal resistance to macrolides was 21%; in approximately the same period (1993-2016) resistance in GAS was 4.9%. This is also evident within a single country: if we analyze the situation in countries like USA, Spain or Portugal, differences in resistance rates between streptococcal species are evident, and the authors should further discuss and analyze these differences. Additionally, within the same species, there are differences in macrolide resistance rates in children and adults (table S1) and this should also be discussed.
Line 454: authors reported to a paper showing positive association between the use of macrolides in food-producing animals and macrolide resistance in S. pneumoniae isolates of human origin. However, the cited paper concerns an ecological analysis that acknowledges that other factors, such as “pneumococcal vaccination coverage, excessive consumption of other classes of antimicrobials, weak antimicrobial stewardship, travel by humans and variations in environmental temperatures”, suggesting that “in vitro and individual level studies are required to further evaluate this hypothesis”. From my point of view, this association needs to be in fact further studied. Also, if animal pneumococcal infections are not very common, the authors should clarify the use of this hypothesis.
Reviewer 2 Report
This review by Berbel et al provides a comprehensive overview of Macrolide resistance (MR) in various species of Streptococci in different geographic regions. The review is well-written and easy to follow. I think the publication of this work will provide insight into the distribution and resistance dynamics of MR in Streptococci sp. I have some minor comments.
1. There are interesting information on the association of different phenotypes/clonal complexes/sequence types to genetic determinants such as resistance genes, mutations, or mobile genetic elements. It would be better if this information can be visualized in table form, at least with the predominant ones. However, I will understand that if authors think it may be difficult to organize.
2. Is Media (%) in the table heading àMedian (%)?
3. In table 1, it was advised to look at the supplementary table for references. I recommend including references in table 1 so that it is easy to understand the data source. If possible, including countries of the corresponding region will be helpful.
Reviewer 3 Report
The article raises one of the most important topics of both modern medicine and agriculture, between which there is a close relationship. This connection is due to the widespread use of antibacterial drugs in both areas and the emergence of resistance to them in bacteria, including multiple resistance to antibiotics, in particular, to macrolides. Despite the relative brevity and conciseness of the review, the authors carried out a tremendous amount of work to study the sources of literature and summarize data on antibiotic resistance in different regions of the world into single tables. In addition, both an analysis of the data available in the world literature and their synthesis were carried out. The article is well written and easy to read. After reading, a general picture of the existing problem is created, and you can try to look for ways to solve it. Indeed, a prudent reduction in the use of antimicrobials is needed where possible, and the search for new targets and the development of drug substances based on different principles, in order to try to avoid the adaptation of microbes to changing conditions. Although, as practice shows, living matter has an enormous potential for variability. I believe that the article does not have significant shortcomings, and those small flaws that are always present in scientific texts when they are submitted for publication can be corrected in the process of proofreading.
Round 2
Reviewer 1 Report
Thank you for the opportunity to review the paper entitled "An overview of macrolide resistance in Streptococci: prevalence, mobile elements and dynamics.
Although some of the questions raised by me in the first revision were addressed by the authors, there are still some aspects that could benefit from an improvement.
My major concern is table S1. The authors stated that they have eliminated data from Farrel et al. when data from other studies was available. However, when performing a literature search, for some countries, there are national studies performed, which would provide more accurate data. Examples of this are Greece (Torumkuney et al., 2018, “Results from the survey of antibiotic resistance (SOAR) 2014-2016 in Greece, JAC), Austria (Paulke-Korinek et al., 2014, “Characteristics of invasive pneumococcal disease in hospitalized children in Austria”, EJP), or Portugal (Silva-Costa et al., 2021, “Pediatric invasive pneumococcal disease three years after PCV13 introduction in the national immunization plan - The continued importance of serotype 3”, Microorganisms).
Moreover, when performing this Pubmed search, I detected another error in the table. In a study from Sweden, (S. pneumoniae, line 109) the authors indicated a macrolide resistance rate of 0.3%, but after reading the original paper, as well as the supplemental material, it was evident that this rate was 8% (Table 2 in the paper indicates percentages and not absolute numbers, so they report 8% resistance and not 8 resistant isolates).
Given this, it is important that the authors confirm all the data presented in the table.
All the lines should have only one reference, so in the case of line 7 (S. pyogenes, the authors should keep the appropriate reference).
The authors accepted my comment regarding the question of S. agalactiae being part of the human microbiota (line 34). However, as it is now it makes no sense, because the authors stated “these streptococci are not usually part of the human microbiota” and “these” refers to the species enumerated in the previous sentence. So one suggestion could be: “Some streptococcal species can cause zoonotic species in the process of animal and human interactions, such as S. suis.”
In table 1, some things changed since the last version. I believe that what the authors calculated was the mean (I understand this correction), but some values changed and I don’t understand why.
In figure 1, we only see S. pneumoniae and S. pyogenes. Why not all the species?
Macrolide consumption: I believe that the question regarding the influence of clonal composition in macrolide resistance should be further explored, providing examples of countries with low macrolide resistance, despite high macrolide consumption. The sentence introduced in lines 398-399 is, from my point of view, insufficient.
The explanation that the authors provided, in response to my comments, regarding differences in macrolide rates among different species should be somehow included in the text. I believe that the inclusion of the possible reasons for this difference, well explained by the authors, would further improve the paper.